# The 2EF Antibody Targets a Unique N-Terminal Epitope of Trop-2 and Enhances the In Vivo Activity of the Cancer-Selective 2G10 Antibody

**DOI:** 10.3390/cancers15143721

**Published:** 2023-07-22

**Authors:** Emanuela Guerra, Marco Trerotola, Valeria Relli, Rossano Lattanzio, Martina Ceci, Khouloud Boujnah, Ludovica Pantalone, Roberta Di Pietro, Manuela Iezzi, Nicola Tinari, Saverio Alberti

**Affiliations:** 1Laboratory of Cancer Pathology, Center for Advanced Studies and Technology (CAST), G. d’Annunzio University of Chieti-Pescara, 66100 Chieti, Italy; emanuela.guerra@unich.it (E.G.); marco.trerotola@unich.it (M.T.); valeria.relli@unibo.it (V.R.); rossano.lattanzio@unich.it (R.L.); martina.ceci@unich.it (M.C.); p.ludovica98@gmail.com (L.P.); 2Department of Medical, Oral and Biotechnological Sciences, G. d’Annunzio University of Chieti-Pescara, 66100 Chieti, Italy; ntinari@unich.it; 3Unit of Medical Genetics, Department of Biomedical Sciences—BIOMORF, University of Messina, 98125 Messina, Italy; khouloud_boujnah@yahoo.com; 4Department of Medicine and Aging Sciences, Section of Biomorphology, G. d’Annunzio University of Chieti-Pescara, 66100 Chieti, Italy; r.dipietro@unich.it; 5Sbarro Institute for Cancer Research and Molecular Medicine, Center for Biotechnology, Department of Biology, College of Science and Technology, Temple University, Philadelphia, PA 19122, USA; 6Department of Neurosciences, Imaging and Clinical Sciences, Center for Advanced Studies and Technnology (CAST), G. d’Annunzio University of Chieti-Pescara, 66100 Chieti, Italy; m.iezzi@unich.it

**Keywords:** Trop-2, cancer vulnerability, precision medicine, target epitopes

## Abstract

**Simple Summary:**

We sought recognition of Trop-2 within difficult-to-reach, densely packed tumor sites. The 2EF mAb was developed, and demonstrated efficient access to Trop-2 at cell–cell junctions in breast cancer cells in culture and in prostate tumors that were not accessible to benchmark anti-Trop-2 antibodies. The 2EF antibody was shown to inhibit the growth of cancer cells in vitro, with the highest activity at high cell density. Correspondingly, 2EF showed the highest anticancer activity on densely packed, Trop-2-expressing tumors. The 2EF mAb enhances the activity of the cancer-only binding 2G10 mAb against tumors in vivo, opening novel avenues for Trop-2-targeted therapy.

**Abstract:**

Trop-2 proteolytic processing in cancer cells exposes epitopes that were specifically targeted by the 2G10 antibody. We sought additional recognition of Trop-2 within difficult-to-reach, densely packed tumor sites. Trop-2 deletion mutants were employed in immunization and screening procedures, and these led to the recognition of a novel epitope in the N-terminal region of Trop-2, by the 2EF antibody. The 2EF mAb was shown to bind Trop-2 at cell–cell junctions in MCF-7 breast cancer cells, and in deeply seated sites in prostate cancer, that were inaccessible to benchmark anti-Trop-2 antibodies. The 2EF antibody was shown to inhibit the growth of HT29 colon tumor cells in vitro, with the highest activity at high cell density. In vivo, 2EF showed anticancer activity against SKOv3 ovarian, Colo205, HT29, HCT116 colon and DU-145 prostate tumors, with the highest impact on densely packed tumor sites, whereby 2EF outcompeted benchmark anti-Trop-2 antibodies. Given the different recognition modes of Trop-2 by 2EF and 2G10, we hypothesized the effective interaction of the two mAb in vivo. The 2EF mAb was indeed demonstrated to enhance the activity of 2G10 against tumor xenotransplants, opening novel avenues for Trop-2-targeted therapy. We humanized 2EF by state-of-the-art CDR grafting/re-modeling, yielding the Hu2EF for therapy of Trop-2-expressing tumors in patients.

## 1. Introduction

Trop-2 is a type-I transmembrane protein, encoded by the tumor-associated calcium signal transducer 2 (*TACSTD2*) gene [1,2,3], a retrotransposon of the *TROP1/TACSTD1/EPCAM* gene [2,4]. Trop-2 drives Ca^2+^, PKC and AKT signaling [5,6,7], and induces tumor cell growth [7,8,9] and metastatic diffusion [9,10]. Association of up-regulation of Trop-2 with poor prognosis of pancreatic, gastric, ovarian, lung and colorectal cancers [10,11] supports a pivotal role in tumor progression [8,9,12].

Sacituzumab Govitecan (SG) (TRODELVY, IMMU-132) was approved by the FDA for therapy in patients with metastatic estrogen-receptor positive and triple-negative breast cancer [13,14] and urothelial carcinomas [15]. However, limited efficacy versus difficult-to-manage toxicity, due to the broad expression of Trop-2 in normal tissues [16], severely plagued further development of anti-Trop-2-targeted antibody (Ab) therapies [17,18,19]. We discovered that activation of Trop-2 for induction of tumor progression requires proteolytic activation by ADAM10 [9,10]. This triggers cancer cell growth and metastatic diffusion, via inhibition of E-cadherin-mediated cell–cell adhesion [9,10]. ADAM10 cleavage occurs in most tumors, including colon, breast and prostate cancers, whereas no Trop-2 cleavage was detected in normal human tissues [9,10].

This led to the generation of the 2G10 monoclonal antibody (mAb), which selectively recognizes cleaved Trop-2 in transformed cells [20,21,22,23]. However, Trop-2 molecules form dimers and multimers at cell–cell junctions [24,25,26], which may hamper Ab binding in tightly packed tumor cell masses. We thus went on to search for mAb that could efficiently bind Trop-2 in densely arrayed cancer sites. Trop-2 deletion mutants were utilized in immunization and screening procedures, and these led to the isolation of the 2EF mAb, that targets a novel epitope in the Trop-2 N-terminal region.

The 2EF mAb was shown to access Trop-2 at cell–cell junctions in breast MCF-7 cancer cells in culture and at deeply seated sites in prostate cancer, that were inaccessible to benchmark anti-Trop-2 mAb. The 2EF mAb diminished HT29 colon tumor cell growth in vitro, with the highest activity being shown on cells growing at high density. In vivo, 2EF inhibited the growth of Trop-2-expressing SKOv3 ovarian, Colo205, HT29, HCT116 colon and DU-145 prostate xenotransplants. The highest anticancer impact was reached on densely packed, established tumors, rather than on isolated tumor cells, whereby 2EF outcompeted benchmark anti-Trop-2 mAbs. We humanized 2EF by state-of-the-art complementarity-determining regions (CDR) grafting/remodeling, yielding Hu2EF. The 2EF mAb was shown to enhance the activity of the cancer-specific 2G10 [20], opening novel avenues for Trop-2-targeted anticancer therapy.

## 2. Materials and Methods

### 2.1. Generation of the Anti-Trop-2 Murine mAb

Deletion mutants of the Trop-2 extracellular domain were used as immunogens in Balb/c mice and for screening of hybridoma clones by ELISA and flow cytometry [21,22]. Differential binding against Trop-2 deletion mutants allowed for the isolation of the N-terminal region-targeting 2EF mAb. The anti-Trop-2 AbT16 [27,28] and the irrelevant mAb p181Bg were utilized as Trop-2-binding versus Trop-2-non-binding control mAb, respectively.

### 2.2. DNA Transfection

Cell transfection and stable transfectant selection in G-418-containing medium were conducted essentially as described [21,22,29].

### 2.3. ELISA

Microtiter plates (Cat. No. 655001, Greiner Bio One, Monroe, NC, USA) were coated overnight at 4 °C with 100 μL/well of 1 μg/mL or 0.1 μg/mL recombinant human Trop-2-IgFc chimera protein (rhTROP-2; Cat. No. 650-T2-100, R&D Systems, Minneapolis, MN, USA), in 0.2 M sodium carbonate buffer (pH 9.4). Well surfaces were blocked with 300 µL/well of blocking buffer (2% skim milk in phosphate Buffer Saline (PBS), 0.05% Tween-20), for 30 min at room temperature (RT). Antibodies were added to the plates at serial 3-fold dilutions, starting from 5–10 μg/mL, 100 µL/well. After incubation for 1 h at RT, Ab binding was detected with 100 µL/well of a 1:2000 dilution of goat anti-human κ-HRP (Cat. No. 2060-05, SouthernBiotech, Birmingham, AL, USA) in blocking buffer. After incubation for 30 min at RT, HRP activity was revealed with 100 μL/well ABTS substrate (AMRESCO, Solon, OH, USA), activated with 20 μL 30% H_2_O_2_ per 10 mL ABTS. The reaction was stopped with 100 µL per well 2% oxalic acid. Absorbance was read at 405 nm [30].

### 2.4. Flow Cytometry

Fluorescence analysis and cell sorting were performed, as described [31], on fluorescence-activated cell analyzers and sorters (FACS Aria III, Canto II, Becton Dickinson, Sunnyvale, CA, USA). mAb were conjugated to Alexa488, 546 or 633 (Life Technologies, Waltham, MA, USA) for direct cell staining. To improve signal-to-noise ratios and the detection of transfectants stained with Alexa488-mAb, subtraction of cell autofluorescence and displacement of Alexa488-stained cells in the red channel were performed essentially as described [32,33].

### 2.5. Immunofluorescence and Confocal Microscopy

Cells grown on glass coverslips were fixed with 4% paraformaldehyde in PBS for 20 min. Permeabilization and blocking were performed in a medium with 10% Fetal Bovine Serum (FBS) and 0.1% saponin. Live cells on glass coverslips were stained in medium with 10% FBS at 37 °C for 5 min and fixed after staining. Slides were analyzed by immunofluorescence (IF) with LSM-510 META and LSM800 (Zeiss, Oberkochen, Germany) confocal microscopes. Three laser beams were used, emitting at wavelengths of 488 nm (argon ion laser, 200 mW, 2–5% applied laser power), 543 nm (diode laser, 1 mW, 50–100% applied laser power), and 633 nm (diode laser, 5 mW, 50% applied laser power). HFT 488/543 or 488/543/633 beam-splitters were used, as needed for multi-color fluorochrome-conjugated Ab analysis, with band-pass emission filters 505 nm to 550 nm (green channel); long-pass 560 nm (for two-color analysis), or band-pass 560 nm to 615 nm (for three-color analyses) in the orange channel; long-pass 650 nm in the deep-red channel. Images were acquired in Multiplex mode, i.e., via sequential acquisition of individual laser lines/fluorescence channels, to prevent cross-channel fluorescence spill-over. Detector gains of ≤770 V were applied to minimize electronic noise. Amplifier gains were ≤2.8. Images were acquired in 1024 × 1024 pixel format, except where indicated. Images were captured as averages of four sequential acquisitions, using 40×/1.2, 63×/1.4 oil DIC objectives (Plan-Apochromat; Zeiss) [20]. Densitometric analysis of independent channel acquisition was performed on representative samples of Trop-2-expressing prostate cancer.

### 2.6. Cloning of the VH and VL Regions of Mouse 2EF

Mouse 2EF-22.21 (2EF) hybridoma cells were generated as described [21,22]. Total RNA was extracted using TRIzol (Invitrogen, Waltham, MA, USA). VH and VL full-length cDNAs were synthesized via nested RT-PCR, using the SMARTer RACE cDNA Amplification Kit (Clontech, Mountain View, CA, USA) with the following primers:2EF VH5′ primer    2EF-H5: 5′-TACACCTTCACTAACTACTGG-3′3′ primers  2EF-H3: 5′-CCCAGTTCCTCTGCACAG-3′                    MCG2b: 5′-GCCAGTGGATAGACTGATGG-3’2EF VL5′ primer   2EF-L5: 5′-AGCCAAAGTGTCAGTACATC-3′3′ primers  JNT319: 5′-CTCCCTCTAACACTCATTCCTGTTGAAGC-3′                   2EF-L3: 5′-GAATCTCCCGACTGTGCTG-3′

The consensus cDNA sequences of 2EF VH and VL are as described (PCT WO201608765). The signal peptide and CDR sequences were identified as in Kabat et al. [34].

### 2.7. Construction of the Chimeric 2EF IgG1/k Ab

The 2EF VH and VL exons were cloned between the SpeI and HindIII sites (for VH) or the NheI and EcoRI sites (for VL) of a mammalian expression vector carrying human g1 and κ constant regions, to generate a chimeric 2EF IgG1/κ Ab (Ch2EF). Genes encoding 2EF VH and VL were synthesized as individual exons, with splice donor signals at the 3′end of the coding region, that were derived from the mouse germline JH3 and Jκ2 sequences, respectively.

### 2.8. Strategy for 2EF Humanization

The VH sequence of the human M17751 cDNA (M17751 VH) [35] and L02325 cDNA (L02325 VH) [36] were chosen as acceptors for humanization. The human VH CDRs were replaced with the murine 2EF VH CDRs, together with selected framework amino acids that were mutated into their murine counterparts. The amino acid sequences of the resulting humanized VH, Hu2EF VH4 and Hu2EF VH5, respectively, are as in PCT WO201608765.

The Vκ region of the human Z46622 cDNA (Z46622 VL) was chosen as acceptor for humanization. The human VL CDRs were replaced with the murine 2EF VL CDRs, together with selected framework amino acids that were mutated into their murine counterparts. The amino acid sequence of the resulting humanized Hu2EF VL1 is as in PCT WO201608765.

The addition of the R38K mutation to Hu2EF VH4 and VH5 led to the generation of Hu2EF VH7 and VH6, respectively. The VL-M4L and VL-Q100G mutations were combined into the Hu2EF VL1 to generate Hu2EF VL2. The pHu2EF-7 vector that expressed the VH7 and VL2 genes was engineered to carry a puromycin resistance (pHu2EF-7-puro).

### 2.9. Generation of NS-0 and YB2/0 Producer Cell Lines

The expression vectors pCh2EF, pHu2EF-4 and pHu2EF-5 were introduced into the NS-0 mouse myeloma (European Collection of Animal Cell Cultures, Salisbury, UK) and in the YB2/0 fucosylation-low rat myeloma [37] (ATCC, Manassas, VA, USA) as described [31].

Expression vectors carrying Hu2EF-4 and Hu2EF-5 variants were transfected into HEK293 cells for transient expression [31]. Antigen binding of transiently expressed antibodies to Trop-2 was analyzed by ELISA in microtiter plates coated with 0.1 μg/mL of rhTrop2 and by flow cytometry analysis of *TROP2* MTE4-14 transfectants.

### 2.10. Expression and Purification of Hu2EF-7

pHu2EF-7-puro stable transfectants in CHO-K1 (ATCC, Manassas, VA, USA) and YB2/0 [37] cells were generated by electroporation. The authenticity of the heavy and light chains produced in CHO-K1-Hu2EF-7 2A2.2 and YB2/0-Hu2EF-7 2D3 cells was confirmed by cDNA sequencing.

### 2.11. Ab-dependent Cellular Cytotoxicity (ADCC) Assay

Jurkat NF-κB/NFAT (ADCC Reporter Bioassay, Core Kit, Cat. No. G7010, Promega, Madison, WI, USA) were utilized as effector/reporter cells. MCF-7 breast cancer cells were used as targets. Briefly, MCF-7 target cells (12,500 cells/well) were seeded in a white, flat-bottom 96-well assay plate (Cat. No. 655001, Corning, Corning, NY, USA) in 100 µL RPMI medium with 10% FBS and 1% Penicillin/Streptomycin (P/S). The day after, 4 µg of purified Hu2EF mAb was serially diluted in ADCC assay buffer (96% RPMI 1640 with L-glutamine, 4% low IgG FBS). Three non-clustered replica wells were utilized for each mAb dilution. One series of wells did not receive the tested mAb (negative control). Jurkat effector cells were added to each well (75,000 cells/well in 25 µL of ADCC assay buffer; effector to target cell ratio = 3:1). After 17 h at 37 °C, 5% CO_2_ 75 µL of Bio-Glo Luciferase Assay Reagent (Cat. No. G7941, Promega) were added to each well and luminiscence was read in a Veritas microplate luminometer (Turner Biosystems, Sunnyvale, CA, USA). Luminescence values were plotted against the mAb concentration Log10 [38]. Data were fitted to a 4-parameter logistic non-linear regression model. The effective 50% concentration (EC_50_) was calculated with GraphPad Prism.

### 2.12. Experimental Tumors

Tumor cell lines and *TROP2* transfectants [29] were injected subcutaneously (5–10 × 10^6^ cells) into 8-week-old female athymic CD1-Foxn1 nu/nu mice (Charles River Laboratories, Calco, Lecco, Italy). The tumor longest/shortest diameters (D/d) were measured every 5–7 days. Tumor volumes were calculated as for an ellipsoid (Dxd^2^/2) [10]. Unless indicated, treatment with anti-Trop-2 or irrelevant (p181bg) mAb was performed by weekly intraperitoneal administration of 30 mg/kg of Ab in PBS, for 4 weeks starting from the day of the inoculation or when the tumors reached 100 mm^3^ of volume.

### 2.13. Study Approval

Procedures involving animals and their care were conducted in compliance with institutional guidelines, national laws and international protocols (D.L. No. 116, G.U., Suppl. 40, 18 February 1992; No. 8, G.U., July 1994; UKCCCR Guidelines for the Welfare of Animals in Experimental Neoplasia; EEC Council Directive 86/609, OJ L 358. 1, 12 December 1987; Guide for the Care and Use of Laboratory Animals, United States National Research Council, 1996), following approval by the Italian Ministry of Health (n° 723/2015-PR) and by the Animal Protection Committee of the Beijing Experimental Animal Center (Research Proposal Approval, 30 June 2015).

### 2.14. Statistical Analysis

One-tailed *t* test was used for the comparison of matrices of Ab-binding densitometry values in confocal microscopy images. EC_50_ values in ADCC activity curves were calculated from dose-response data fitted to a 4-parameter-logistic non-linear regression model. ANOVA [39] and *t* test implementing a *post-hoc* Bonferroni correction were used to comparatively assess tumor growth curves. Data were analyzed using Sigma Stat 4.0 (SPSS Science Software UK Ltd., Chicago, IL, USA) and GraphPad Prism 7 (GraphPad Software Inc., La Jolla, CA, USA).

## 3. Results

We discovered that activation of Trop-2 as a driver of tumor progression requires proteolytic cleavage by ADAM10 [9,10]. This led to the development of the 2G10 mAb, which selectively recognizes ADAM10-cleaved Trop-2 in transformed cells [21,22,23]. However, Trop-2 molecules form dimers and higher-order multimers at cell–cell contacts [24,25,26]. This may hamper Ab binding in difficult-to-penetrate tumor cell masses. We thus searched for mAb that could efficiently bind Trop-2 at cell–cell junctions in densely arrayed cancer sites.

### 3.1. Generation of mAbs Directed against Accessible Trop-2 Sites in Tightly Packed Tumor Cells

We designed recognition strategies for distinct regions of Trop-2 to generate mAb with selective reactivity against the N-terminal region of Trop-2, i.e., at a distant site from the 2G10 target region and from the Trop-2 immunodominant epitope [19,28,40]. Trop-2 deletion mutants were utilized for hybridoma screening and epitope mapping [21,22], by ELISA, flow cytometry and fluorescence microscopy on cleaved/activated or wild type (wt) recombinant Trop-2. This and the lack of competition versus the 2G10 mAb family showed that the 2EF mAb recognizes a distinct target epitope in the Trop-2 N-terminus.

### 3.2. Effective Binding of 2EF to Tumor Cells at Deeply Seated Cancer Sites

We went on to assess whether 2EF can effectively recognize Trop-2 in difficult-to-reach regions in densely packed tumor cells. IF staining of MTE4-14/Trop-2 transfectants showed efficient recognition of Trop-2 by 2EF (Figure 1A). We then showed that 2EF gains access to Trop-2 at cell–cell junctions in breast MCF-7, which grow in culture as tightly packed cell islands. Much less efficient binding of Trop-2 at cell junctions was shown by 2G10 and by the benchmark AbT16 mAb (Figure 1B).

We then assessed whether 2EF could gain access to Trop-2 at deeply seated tumor sites, that were inaccessible to benchmark anti-Trop-2 mAb. Multiplex confocal microscopy IF analysis of prostate cancer revealed deeper penetration/effective binding of 2EF to cancer cells, as opposed to 2G10 and to the immunodominant epitope-binding AbT16 (Figure 2).

Comparative 2EF and 2G10 binding were assessed with Fiji/ImageJ 2.9.0. Normalized pixel values were obtained in regions of interest (ROI) of the tumor. A threshold of differential intensity of 2EF versus 2G10 signals was identified as the divergence point of binding curves versus depth of penetration in tumor islands (Appendix A). In other words, we determined the threshold distance from the tumor stroma whereby the 2G10 signal started to fall, whereas that of 2EF remained high. This was computed to be 35.6 ± 6.0 µm (mean ± SEM; range: 10–100 µm; *n* = 15). The largest differential intensity/deepest penetration of 2EF versus 2G10 was shown to be 61.6 ± 8.2 µm (mean ± SEM; range: 21.5–140.0 µm; *p* = 0.0102). The ratio of the areas-under-the-curve of 2G10 versus 2EF in ROI was 55 ± 0.03% (mean ± SEM; range: 44.8–73.3%; *p* = 0.0028), indicating an almost two-fold increase in efficiency in penetration/binding of 2EF at central tumor areas. Notably, no Ab penetration/staining was detectable in the lumen of the prostate cancer gland-like structure (Appendix A).

Grade 2 polynomial curves were utilized to fit the density sequences of 2G10 and 2EF signals in ROI (Appendix A). A comparison of curve parameters (y, x^2^ and R^2^) confirmed the higher absolute values of 2EF binding versus 2G10 at deeply seated sites in prostate cancer. Moreover, in essentially all cases, 2G10 binding curves showed a higher curvature/larger differential binding versus 2EF, which showed a smoother data distribution/higher homogeneity in staining patterns (Appendix A).

### 3.3. Engineering of Chimeric and Humanized 2EF

To prevent eliciting a human anti-mouse Ab response upon administration to patients [41], the 2EF mAb was humanized. Cloned 2EF VH and VL exons were transferred into a mammalian expression vector carrying human g1 and κ constant regions to generate the chimeric 2EF IgG1/κ Ab (Ch2EF).

The VH sequence encoded by the human M17751 cDNA (M17751 VH) [35] was chosen as an acceptor for CDR grafting. At framework positions 44, 48, 67, 69 and 71, where the three-dimensional model of the 2EF variable regions suggested significant contact with the CDRs, human amino acids were reverted to the corresponding murine residues, to generate the Hu2EF-4 construct. A parallel humanized VH (Hu2EF VH5) was designed using the VH sequence encoded by the human L02325 cDNA (L02325 VH) [36]. At framework positions 48, 67, 69 and 71, human amino acid residues were substituted by the corresponding mouse residues to prevent a steric clash with CDR regions.

The human Vκ region encoded by the Z46622 cDNA (Z46622 VL) was chosen as an acceptor for CDR grafting. At position 49, where the three-dimensional model of the 2EF variable regions indicated significant contact with the CDRs, the human amino acid was substituted with the corresponding murine residue.

The Hu2EF-4 was constructed that comprised the humanized Hu2EF VH4 and VL1; Hu2EF-5 comprised VH5 and VL1. Additional VH mutants at position 38 (VH-R38K), 40 (VH-A40R), and 43 (VH-Q43H), and VL mutants at position 4 (VL-M4L) and 100 (VL-Q100G) were generated. Each of these additional variants was combined with the unmodified humanized VL (or VH) genes in mammalian expression vectors (Appendix A). The VH-R38K improved Trop-2 binding (Figure 3). The Hu2EF VH4 carrying the R38K mutation was named Hu2EF-VH7, and the Hu2EF VH5 carrying the R38K was named Hu2EF-VH6. As to the humanized VL gene, both VL-M4L and VL-Q100G slightly improved Trop-2 binding (Figure 3). These two mutations were combined in Hu2EF VL1 to generate Hu2EF VL2.

### 3.4. Hu2EF-7 Binding to Trop-2

The binding affinity of Ch2EF, Hu2EF-6 and Hu2EF-7 to recombinant Trop-2 was assessed by ELISA and competition flow cytometry. Independent Trop-2-expressing cell lines, i.e., KM12SM/Trop-2, Colo205, HT29, DU-145 and MCF-7 cells, were incubated with Alexa488-labeled mouse 2EF and serial amounts of unlabeled mouse 2EF, Hu2EF-6 or Hu2EF-7. Competition profiles were shown to be similar across different cancer cell lines (Figure 4) and MTE4-14 transfectants. A small but detectable improvement of competition efficiency was observed for Hu2EF-7 over Hu2EF-6 (Figure 4, MCF-7 panel), indicating Hu2EF-7 as an efficient humanized counterpart of mouse 2EF.

The effector functions of IgG are dependent on the glycosylation of the Fc region [42], and low-fucose IgG1 exhibits a higher ADCC activity compared to highly fucosylated IgG1 [43]. Hence, we expressed Hu2EF-7 in the fucosylation-low YB2/0 rat myeloma [37]. The binding efficiency of the YB2/0 Hu2EF-7 was assessed against KM12SM transfectants [9,10], expressing wtTrop-2 or mutagenized Trop-2 devoid of glycosylation or resistant to proteolysis. Benchmarks were the mouse 2EF-22.21-Alexa488 and Hu2EF-7-Alexa488 produced in CHO-K1 cells. In all cases, fucosylation-low Hu2EF-7 efficiently bound Trop-2 targets (Appendix A), suggesting this as a viable means for therapeutic development in patients.

### 3.5. Ab-Dependent Cell-Mediated Cytotoxicity by Hu2EF

ADCC can mediate a considerable fraction of mAb anticancer activity [44]. The capacity of Hu2EF mAb to mediate ADCC was measured using effector Jurkat cells stably expressing the V158 (high affinity) FcγRIIIa receptor and a firefly luciferase reporter gene, driven by an NFAT response element, which becomes activated when the FcγRIIIa receptor is engaged in ADCC. The target MCF-7 breast cancer cells were selected as expressing endogenous Trop-2 at levels corresponding to average amounts in primary human cancers [8] (Figure 5). Different amounts of purified mAb (range 10 pM–1 µM) were added to monolayers of MCF-7 target cells, together with Jurkat effector cells, at the indicated effector/target cell ratios. Efficient lysis of cancer cells by Hu2EF was observed, with a good EC_50_ of 0.3 nM (Figure 5).

### 3.6. Inhibition of Cancer Cell Growth by 2EF

Cell confluency in culture was previously shown to affect Trop-2 proteolysis in transformed cells [9] and activation pathways for cell growth [6]. We thus analyzed the impact of 2EF versus 2G10 on HT29 colon cancer cells growing in culture at high versus low cell density. The 2G10 mAb did inhibit HT29 cell growth better than 2EF at low cell density. However, the 2EF mAb fared as well as 2G10 in high-density cell cultures (Figure 6A,B).

These findings supported a model of high capacity of 2EF to recognize Trop-2 in densely arrayed cancer cells. We went on to explore whether 2EF maintained such capacity in vivo. The therapeutic efficacy of the 2EF versus 2G10 against HT29 human cancer xenografts growing in nude mice was assessed. Tumor-bearing mice were treated either at injection or when tumors reached an average volume of 0.1 cm^3^ (“individual cell” versus “established tumor” models, respectively). Mice in the control group received an irrelevant isotype-matched mAb or the anti-Trop-2 AR47A6.4.2 mAb [45], as an activity reference. The 2EF mAb showed anticancer activity in both models. In the established cancer model 2EF outcompeted the benchmark anti-Trop-2 mAbs (Figure 6C,D).

### 3.7. Xenograft Growth Inhibition by 2EF

We then challenged 2EF activity against other Trop-2-expressing tumors. Significant growth inhibition by 2EF was shown against ovarian cancer SKOv-3 (*p* = 0.0063 by ANOVA test with Bonferroni post-hoc correction versus control mAb), with higher activity than AR47A6.4.2. Corresponding findings were obtained against the colon cancer Colo205 and the metastatic colon cancer HCT-116 U5.5 [9,10] (Figure 7). This demonstrated high anticancer activity of 2EF across tumor models, supporting it as a broadly applicable anticancer reagent.

The 2EF and 2G10 mAb recognize cancer-expressed Trop-2 in a sharply distinct manner. Hence, we hypothesized that they could effectively interact in cancer cell killing. Athymic nude mice were subcutaneously injected with human DU-145 prostate cancer cells. Injected mice were randomized and treated with 30 mg/kg mAb weekly until sacrifice. Mice treated with 2EF plus 2G10 were administered half dose (15 mg/kg) of each mAb weekly until sacrifice. The AR47A6.4.2 and AbT16 anti-Trop-2-immunodominant site mAb were used as benchmarks. Treatment began when tumors reached an average volume of 0.1 cm^3^. Our findings showed the efficacy of both 2EF and 2G10. Remarkably, though, the highest efficacy was revealed in the combination treatment group (Figure 8), suggesting enhancement of 2G10 anticancer activity by 2EF.

We did not observe any adverse effect in treated mice versus controls, as indicated by normal behavior, clinical appearance and lack of variation in body weight during treatment. Correspondingly, we did not observe any adverse effect in treated Cynomolgous monkeys as compared with control primates (manuscript in preparation).

## 4. Discussion

High-potency Trop-2-targeted therapy for advanced cancer is urgently needed [17]. However, next-generation approaches remain hampered by the expression of Trop-2 in normal tissues [6,8,9,10,28,46], including the epidermis, endometrium, esophagus, tonsil, lung, kidney, salivary glands and breast [6,21,22]. Such on-target/off-tumor binding was shown to potentially lead to unmanageable toxicity [17,18]. We discovered that activation of Trop-2 as a driver of tumor progression requires proteolytic cleavage by ADAM10 [9,10]. Trop-2 processing by ADAM10 occurs in most tumors but not in normal human tissues [9,10]. Trop-2 cleavage by ADAM10 exposes a previously inaccessible protein groove in a cancer-specific manner. This region was recognized by the newly developed 2G10 mAb family for exploiting selective cancer vulnerability in patients [20,21,22].

However, Trop-2 molecules form stable dimers [24] and multimers at cancer cell–cell contacts [25,26]. Hence, target epitopes in Trop-2 may become less accessible to Ab binding in tightly packed cancer cell masses. We thus searched for a mAb with improved access to tightly packed cells in culture and in tumor xenotransplants. Through Trop-2 structure-function informed analysis [9,24,25] and deletion mutagenesis-based immunization and screening strategies, we succeeded in generating the 2EF mAb, that was shown to recognize a novel epitope in the N-terminal regions of Trop-2.

Confocal microscopy analysis of 2EF binding to Trop-2 in breast MCF-7 cancer cells, which grow in culture as tightly packed cell islands, showed that 2EF can efficiently bind Trop-2 at cell–cell junctions, at variance with the 2G10 and benchmark AbT16 mAb. Multiplex confocal microscopy analysis of prostate cancer samples correspondingly showed much deeper 2EF penetration in cancer cell islands, as opposed to benchmark immunodominant epitope-binding mAb. Consistently, 2EF showed powerful antitumor activity in multiple Trop-2-expressing preclinical tumor models, among them the SKOv3 ovarian, Colo205, HT29, HCT116 U5.5 colon and DU-145 prostate cancers. Remarkably, the highest anticancer impact was reached in the most difficult-to-treat, densely packed established tumors, rather than on isolated tumor cells, whereby 2EF outcompeted all benchmark anti-Trop-2 mAb that were tested.

In order to obtain a mAb that could be repeatedly administered to patients without eliciting a human anti-mouse Ab response [41], the 2EF mAb was humanized by state-of-the-art CDR grafting/remodeling. Trop-2 binding profiles of Hu2EF-7 versus the Ch2EF, Hu2EF-4, Hu2EF-5 and Hu2EF-6 intermediates, as determined by ELISA and competition flow cytometry, were shown to be similar across distinct cancer cell types, indicating Hu2EF-7 as a successfully humanized form of mouse 2EF.

The effector functions of recombinant therapeutic IgG are dependent on the glycosylation of the Fc region [42]. A strategy to improve binding to the Fc receptor is to modify mAb glycosylation states [47], as low fucose IgG1 exhibit higher in vitro and in vivo ADCC activity compared to highly fucosylated IgG [43]. As both CHO and NS-0 cells possess intrinsic fucosyl-transferase activity, we expressed Hu2EF-7 in YB2/0 cells, a fucosylation-low rat myeloma [37]. Fucosylation-low Hu2EF-7 efficiently bound Trop-2 targets, in a comparable, if not better, manner versus the parental mAb, suggesting this as a viable means for therapeutic approaches in cancer patients.

## 5. Conclusions

Recent findings support the cancer-selective Hu2G10 as instrumental for next-generation anti-Trop-2 ADC [21,22,23]. The efficacy of 2EF against several tumor models, its capacity to reach deeply seated cancer sites and its lack of toxicity in animal models support 2EF as a novel candidate and an efficient anti-Trop-2 therapeutic mAb. The efficient humanization of 2EF makes Hu2EF a candidate for the development of novel anti-Trop-2 ADC. The enhancement of the 2G10 cancer-specific mAb in vivo, and the expected reduction in on-target/off-tumor toxicity pave the way for novel approaches for Trop-2-targeted therapy. It will be equally interesting to assess how this innovative concept may work in combination or in sequence versus other anti-Trop-2 mAb with different target epitopes, e.g., SG.

## Figures and Tables

**Figure 1 cancers-15-03721-f001:**
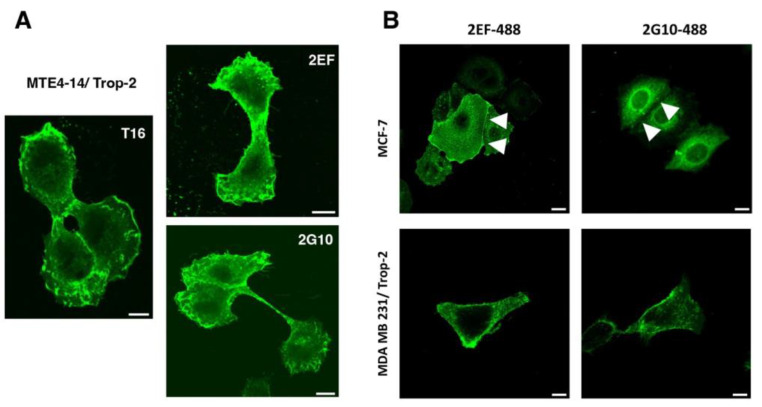
The 2EF binding to Trop-2-expressing cancer cells. (**A**) Recognition of Trop-2 by 2EF-Alexa488 on MTE4-14/Trop-2 transfectants. Confocal microscopy IF analysis of cells cultured in adhesion to substrate was performed with anti-Trop-2 mAb directly conjugated to chromophores. The 2G10-Alexa488 and AbT16-Alexa488 were used as anti-Trop-2 benchmarks. Scale bars, 10 µm. (**B**) IF analysis of MCF-7 epithelioid and MDA-MB231 mesenchymal breast cancer cells. White arrowheads indicate cell–cell junctions with higher differential binding of 2EF versus 2G10. Scale bars, 10 µm.

**Figure 2 cancers-15-03721-f002:**
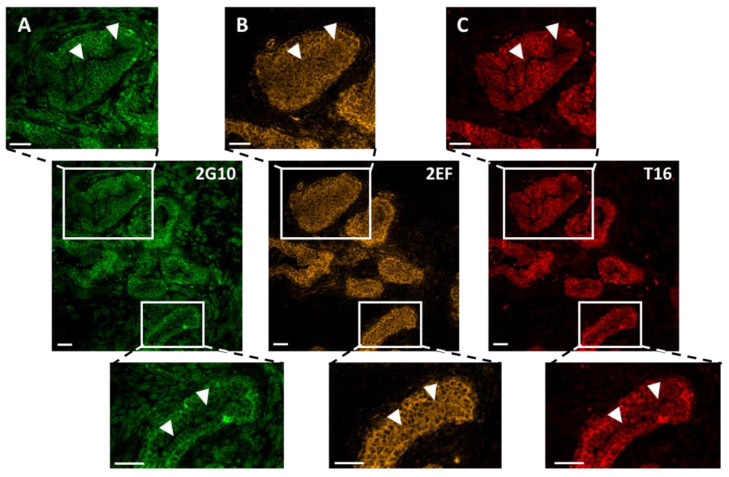
Comparative binding of 2EF, 2G10 and AbT16 to representative samples of Trop-2-expressing prostate cancer by IF confocal microscopy. (**A**) 2G10-Alexa488; (**B**) 2EF-Alexa546; (**C**) AbT16-Alexa633. The three anti-Trop-2 mAb were utilized simultaneously. Individual signals for each fluorophore were acquired independently, in a sequential manner across fluorescence channels. Magnified regions of interest are indicated by white rectangles. White arrowheads indicate the tumor regions with higher differential penetration/binding capacity of 2EF-Alexa488. Scale bars, 50 µm.

**Figure 3 cancers-15-03721-f003:**
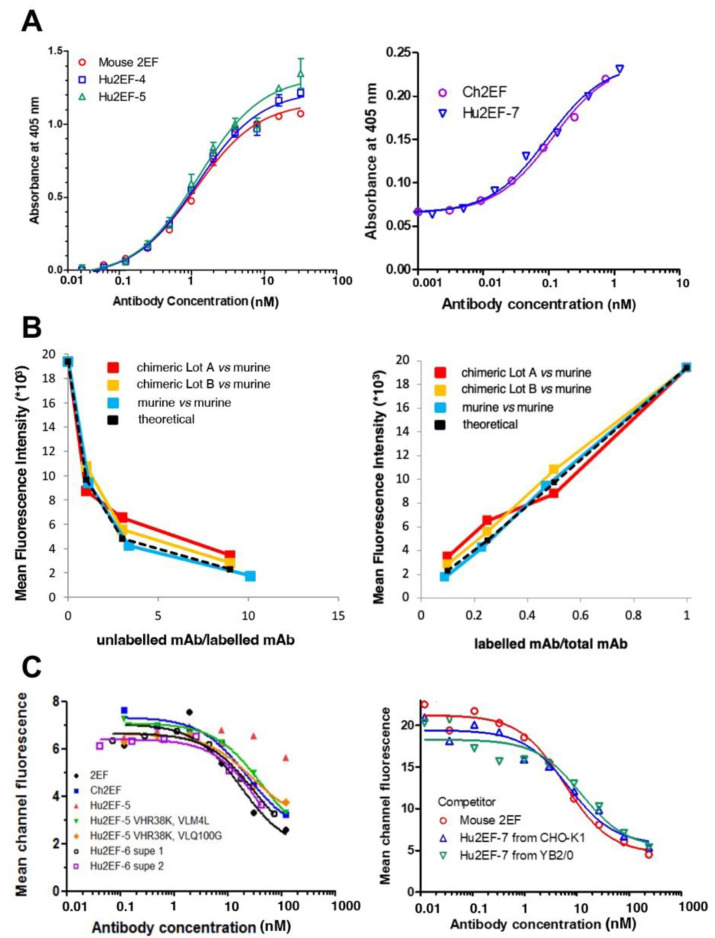
Binding affinities of the engineered chimeric and humanized 2EF. (**A**) ELISA binding assay of Ch2EF (batches A, B), Hu2EF-4, Hu2EF-6, Hu2EF-7 to recombinant Trop-2-Fc chimera immobilized to the substrate. Murine 2EF was used as a benchmark. Trop-2-Fc was used at 1 µg/mL (**left**) or at 0.1 µg/mL (**right**). Absorbance values (Y-axis) are plotted at each tested Ab concentration (X-axis). (**B**) Data from competitive flow cytometry analysis of murine 2EF and Ch2EF binding to MTE4-14/Trop-2 transfectants were plotted as the ratio of unlabeled mAb/labeled mAb versus mean fluorescence intensity (**left**) and as labeled mAb/total mAb ratio versus mean fluorescence intensity (**right**). Theoretical binding curves are in black. (**C**) Data from competitive flow cytometry analysis of Colo205 colon cancer cells stained with murine 2EF-Alexa488, as competed-out by unlabeled engineered 2EF variants: (**left**) Ch2EF, Hu2EF-5, Hu2EF-6 (batches 1 and 2); (**right**) Hu2EF-7 produced in CHO-K1 or YB2/0 low-fucosylation cells.

**Figure 4 cancers-15-03721-f004:**
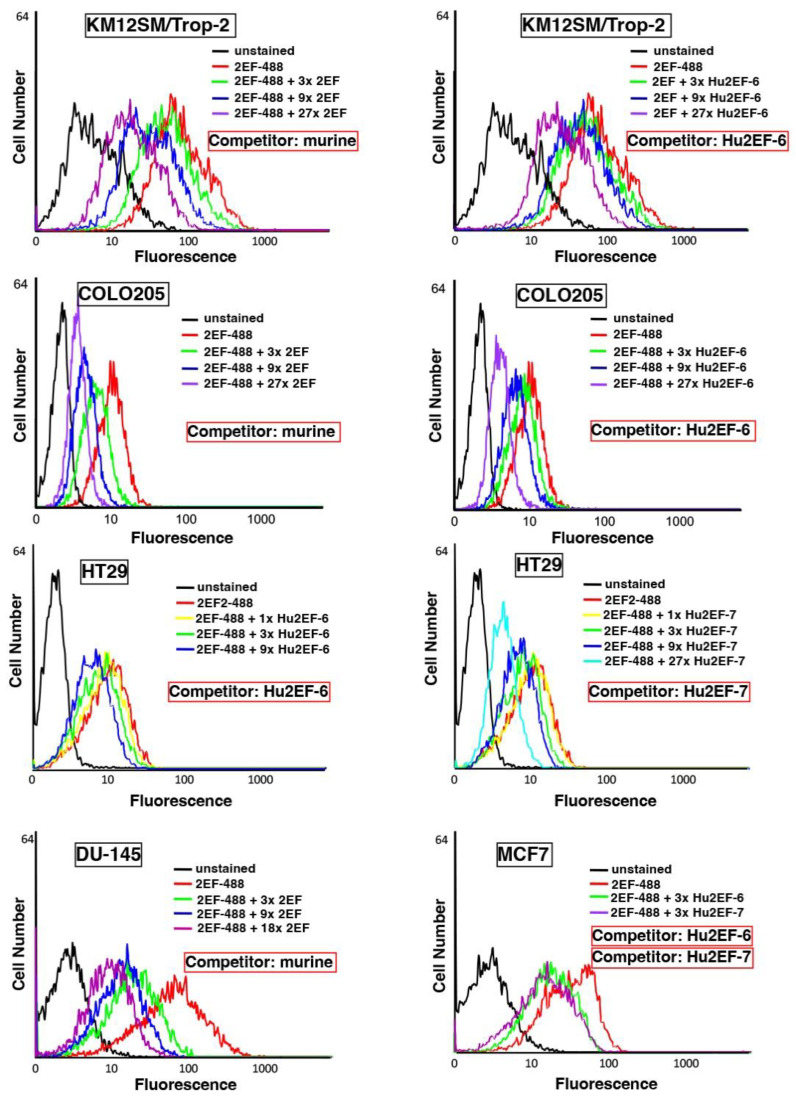
Hu2EF engineered variants binding properties. Flow cytometry analysis of Trop-2-expressing cancer cells, stained with the murine 2EF-Alexa488 mAb, as competed-out by unlabeled murine 2EF (Competitor: murine), unlabeled Hu2EF-6 (Competitor: Hu2EF-6) or unlabeled Hu2EF-7 (Competitor: Hu2EF-7). Competing mAb were added at the indicated ratios (color-coded from 1× to 27×) to fixed amounts of 2EF-Alexa488. Stained cancer cells were, from top to bottom, KM12SM/Trop-2, Colo205, HT29, DU-145 and MCF-7. Relative affinity correlates with the reduction in fluorescence signals. Unstained controls are in black.

**Figure 5 cancers-15-03721-f005:**
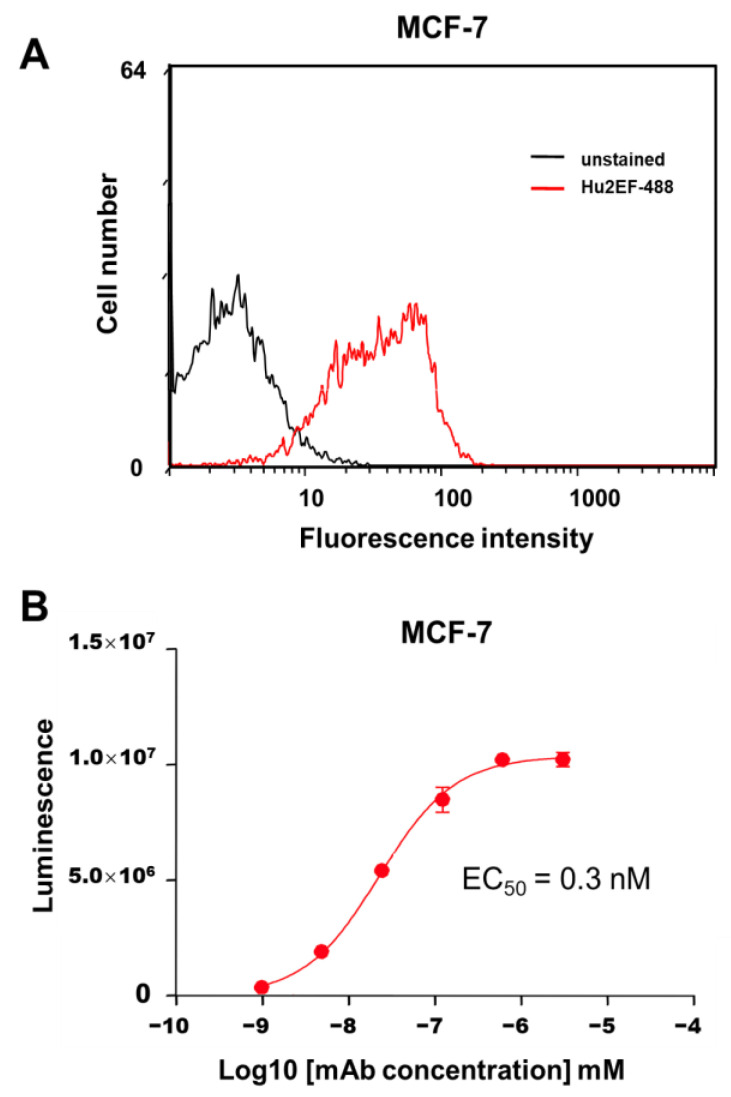
Hu2EF immune-mediated killing of Trop-2-expressing cancer cells. (**A**) Flow cytometry analysis of Hu2EF binding to target MCF-7 cells. (**B**) Hu2EF-mediated ADCC dose-response curve, following incubation of MCF-7 target cells with serial dilutions of the Hu2EF mAb and effector Jurkat NF-kB/NFAT-reporter cells. Luminescence data upon NF-kB/NFAT-reporter activation (three replica wells per data point) were plotted against mAb concentration. Error bars: SEM.

**Figure 6 cancers-15-03721-f006:**
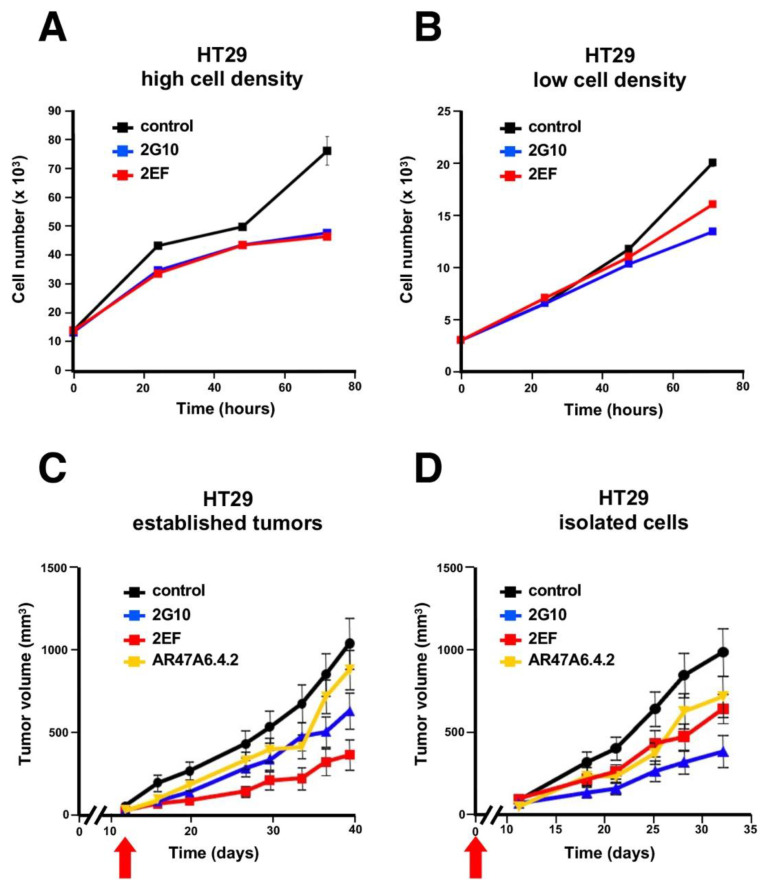
Growth inhibition of Trop-2-expressing HT29 colon cancer cells by 2EF versus 2G10. HT29 cells were seeded in culture at high (**A**) versus low (**B**) cell density. Murine 2EF or 2G10 mAb were added every 24 h. (**C**,**D**) Immunotherapy of HT29 colon cancer xenografts. Injected mice were randomized (*n* = 16 per group) and treated weekly with 30 mg/kg of the 2EF, 2G10 or AR47A6.4.2 mAb until sacrifice. Treatment began when (**C**) tumors reached an average volume of 100 mm^3^ (established tumor) or (**D**) at the time of tumor cell injection (isolated cells), as indicated (red arrows). Error bars: SEM.

**Figure 7 cancers-15-03721-f007:**
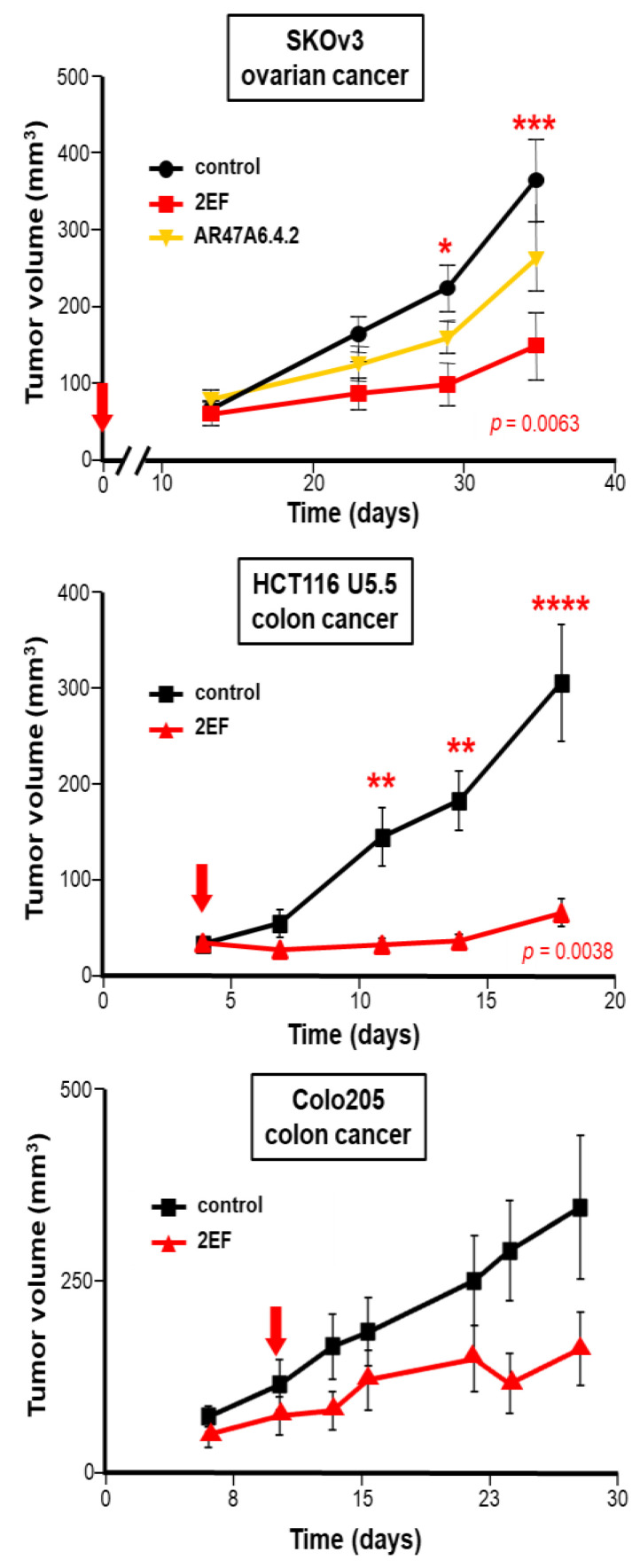
Immunotherapy of Trop-2-expressing human cancer xenografts. Athymic nude mice were subcutaneously injected with the human SKOv3 ovarian, HCT116 U5.5 or Colo205 colon cancer cells. Injected mice were randomized (*n* = 16 per group) and treated with 30 mg/kg 2EF or 2G10 mAb administered weekly until sacrifice. The AR47A6.4.2 was used as a benchmark for anti-Trop-2 immunotherapy. Mice in the control groups received an irrelevant isotype-matched mAb. Red arrows: treatment was started at the time of injection of isolated tumor cells (time = 0) (SKOv3 ovarian cancer, upper panel) or when tumors reached an average volume of 100 mm^3^ (HCT116 U5.5, Colo205 colon cancers). Error bars: SEM. Significantly higher growth inhibition by 2EF versus control mAb was shown by the ANOVA test with Bonferroni post-hoc correction. *: *p* = 0.05; **: *p* = 0.01; ***: *p* = 0.0063, ****: *p* = 0.0038.

**Figure 8 cancers-15-03721-f008:**
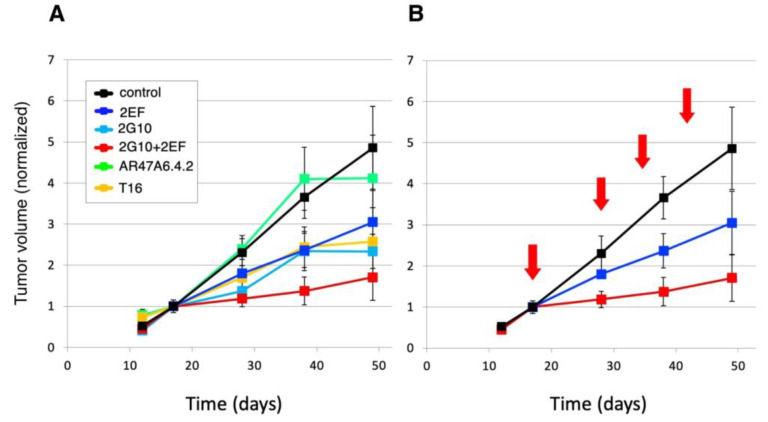
The 2EF mAb enhances the in vivo activity of the 2G10 cancer-specific mAb [20]. Athymic nude mice were subcutaneously injected with human DU-145 prostate cancer cells. Injected mice were randomized (*n* = 16 per group) and treated with 30 mg/kg mAb (2EF, 2G10, 2EF plus 2G10, AR47A6.4.2 or AbT16) weekly until sacrifice. Mice treated with 2EF plus 2G10 were administered 400 µg of each mAb weekly until sacrifice. Treatment began when tumors reached an average volume of 100 mm^3^. (**A**) The AR47A6.4.2 (green) and AbT16 (orange) anti-Trop-2-immunodominant site mAb were used as benchmarks. The 2EF (blue), 2G10 (cyan) and 2EF plus 2G10 (red) treatments are indicated. (**B**) Details of treatment schedules (red arrows) and of the impact of 2EF versus 2EF plus 2G10 on tumor growth are shown. Tumor volumes were normalized versus volume at first treatment. Error bars: SEM.

## Data Availability

All relevant data are included in this article.

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
