# Peer review of "The 2EF Antibody Targets a Unique N-Terminal Epitope of Trop-2 and Enhances the In Vivo Activity of the Cancer-Selective 2G10 Antibody"

_cancers, 2023, doi:10.3390/cancers15143721_

Round 1
Reviewer 1 Report (Previous Reviewer 1)
The article entitled “The 2EF antibody targets a unique N-terminal epitope of Trop-2 and enhances the in vivo activity of the cancer-selective 2G10 antibody” has been evaluated. 2EF antibody was shown to inhibit the growth of HT29 colon tumor cells in vitro, with highest activity at high cell density. In vivo, 2EF showed anticancer activity against SKOv3 ovarian, Colo205, HT29, HCT116 colon and DU-145 prostate tumors, with highest impact on densely-packed tumor sites, whereby 2EF outcompeted benchmark anti-Trop-2 antibodies. The 2EF mAb was indeed demonstrated to enhance the activity of 2G10 against tumor xenotransplants, opening novel avenues for Trop-2-targeted therapy.
The MS was originally submitted in cancers-2306275; based on the reviewer’s suggestion, authors made significant revisions. The MS can be acceptable for publication.
Minor editing of English language required
Reviewer 2 Report (Previous Reviewer 3)
the work is interesting and can be published in its current form
This manuscript is a resubmission of an earlier submission. The following is a list of the peer review reports and author responses from that submission.
Round 1
Reviewer 1 Report
The manuscript entitled “ The 2EF antibody targets a unique epitope of Trop-2 and syner-2 gizes in vivo with the cancer-selective 2G10 antibody” is very well designed and written. This present study authors developed 2G10 monoclonal antibody (mAb), subsequently humanized to Hu2G10, which selectively recognizes Trop-2 in transformed cells. Further they employed in immunization and screening procedures that led to the recognition of a novel epitope in the N-terminal region of Trop-2 by the 2EF mAb. We found that 2EF could access Trop-2 at cell-cell junctions in breast MCF-7 cancer cells in culture and in deeply seated sites in DU-145 prostate tumors, that were inaccessible to benchmark anti-Trop-2 mAb. The 2EF mAb directly inhibited HT29 colon tumor cell growth in vitro, the highest activity being shown on high-density growing cells. In vivo, 2EF inhibited the growth of Trop-2-expressing SKOv3 ovarian, Colo205, HT29, HCT116 colon and DU-145 prostate cancers. The humanized 2EF by state-of-the-art CDR grafting/re modeling, yielding Hu2EF. The 2EF mAb showed synergy with 2G10 in vivo, opening novel avenues for Trop-2-targeted anticancer therapy. Still, I have some concerns about improving the quality of the manuscript. I have mentioned some comments in the manuscript file; if authors give attention to those questions should be welcomed.
Minor
-Manuscript needs both typo checking by authors and English language correction.
-Abbreviations used for the first time should be given in full name (and no need for repetition later in
the text).

Reviewer 2 Report
Authors have described 2EF antibody targets a unique epitope of Trop-2 and synergizes in vivo with the cancer-selective 2G10 antibody. The concept of MS is good. However, the Introduction section can be improved incorporating more knowledge related to the field.
The specific comments, which could help to improve the manuscript, are:
1. Simple Summary and abstract sections look similar. Summary should include key points.
2. The manuscript should be revised for grammatical & punctuation errors.
3. Better to add a list of abbreviations.
4. Section: 2.3. (ELISA) ; 2.5. (Immunofluorescence and confocal microscopy); 2.12. (Antibody-dependent cellular cytotoxicity (ADCC) assay). Provide a suitable reference.
5. Conclude the findings and future directions in an effective way to improve research design. It would be better to add research gap and future prospects related to the topic in conclusion section.
Reviewer 3 Report
In the manuscript entitled " The 2EF antibody targets a unique epitope of Trop-2 and synergizes in vivo with the cancer-selective 2G10 antibody “the authors identify Trop-2 epitopes exposes during proteolytic processing in cancer cells that were specifically targeted by the 2G10 antibody.
I congratulate the authors on their research. The analysis itself is extensive and seems technically adequate
It is well written.
It is methodologically correct
It's interesting
a comment from the authors on the possible synergistic role with sacituzumab govitecan, the potential use in association or in sequence, on the potentially toxicity profile considering that no Trop-2 cleavage was detected in normal human tissues and finally on whether the antibody can be drug conjugate (ADC) would be very interesting.